# Contribution of Second Trimester Sonographic Placental Morphology to Uterine Artery Doppler in the Prediction of Placenta-Mediated Pregnancy Complications

**DOI:** 10.3390/jcm11226759

**Published:** 2022-11-15

**Authors:** Eran Ashwal, Jasmine Ali-Gami, Amir Aviram, Stefania Ronzoni, Elad Mei-Dan, John Kingdom, Nir Melamed

**Affiliations:** 1Division of Maternal-Fetal Medicine, Department of Obstetrics and Gynecology, McMaster University, Hamilton, ON L8N 3Z5, Canada; 2Division of Maternal-Fetal Medicine, Department of Obstetrics and Gynecology, Sunnybrook Health Sciences Centre, University of Toronto, Toronto, ON M4N 3M5, Canada; 3Division of Maternal-Fetal Medicine, Department of Obstetrics and Gynecology, North York General Hospital, University of Toronto, Toronto, ON M2K 1E1, Canada; 4Division of Maternal-Fetal Medicine, Department of Obstetrics and Gynecology, Mount Sinai Hospital, Toronto, ON M5G 1X5, Canada

**Keywords:** placenta, uterine artery Doppler, placental area, placental morphology

## Abstract

Background: Second-trimester uterine artery Doppler is a well-established tool for the prediction of preeclampsia and fetal growth restriction. At delivery, placentas from affected pregnancies may have gross pathologic findings. Some of these features are detectable by ultrasound, but the relative importance of placental morphologic assessment and uterine artery Doppler in mid-pregnancy is presently unclear. Objective: To characterize the association of second-trimester sonographic placental morphology markers with placenta-mediated complications and determine whether these markers are predictive of placental dysfunction independent of uterine artery Doppler. Methods: This was a retrospective cohort study of patients with a singleton pregnancy at high risk of placental complications who underwent a sonographic placental study at mid-gestation (160/7−246/7 weeks’ gestation) in a single tertiary referral center between 2016–2019. The sonographic placental study included assessment of placental dimensions (length, width, and thickness), placental texture appearance, umbilical cord anatomy, and uterine artery Doppler (mean pulsatility index and early diastolic notching). Placental area and volume were calculated based on placental length, width, and thickness. Continuous placental markers were converted to multiples on medians (MoM). The primary outcome was a composite of early-onset preeclampsia and birthweight < 3rd centile. Results: A total of 429 eligible patients were identified during the study period, of whom 45 (10.5%) experienced the primary outcome. The rate of the primary outcome increased progressively with decreasing placental length, width, and area, and increased progressively with increasing mean uterine artery pulsatility index (PI). By contrast, placental thickness followed a U-shaped relationship with the primary outcome. Placental length, width, and area, mean uterine artery PI and bilateral uterine artery notching were all associated with the primary outcome. However, in the adjusted analysis, the association persisted only for placenta area (adjusted odds ratio [aOR] 0.21, 95%-confidence interval [CI] 0.06–0.73) and mean uterine artery PI (aOR 11.71, 95%-CI 3.84–35.72). The area under the ROC curve was highest for mean uterine artery PI (0.80, 95%-CI 0.71–0.89) and was significantly higher than that of placental area (0.67, 95%-CI 0.57–0.76, *p* = 0.44). A model that included both mean uterine artery PI and placental area did not significantly increase the area under the curve (0.82, 95%-CI 0.74–0.90, *p* = 0.255), and was associated with a relatively minor increase in specificity for the primary outcome compared with mean uterine artery PI alone (63% [95%-CI 58–68%] vs. 52% [95%-CI 47–57%]). Conclusion: Placental area is independently associated with the risk of placenta-mediated complications yet, when combined with uterine artery Doppler, did not further improve the prediction of such complications compared with uterine artery Doppler alone.

## 1. Introduction

Placenta-mediated pregnancy complications are a heterogenous group of disorders that include preeclampsia and fetal growth restriction [1,2,3]. As these conditions are associated with significant maternal, fetal, and neonatal mortality and morbidity [4], early prediction of these disorders is a major research priority [5,6].

Second-trimester uterine artery Doppler is a well-established tool for the prediction of preeclampsia and fetal growth restriction, especially in patients at high risk of placental complications [7,8,9]. Other sonographic placental markers, including measures of placental size (length, width, thickness, and volume), placental appearance, and umbilical cord assessment (placental insertion site and number of vessels) have also been shown to be associated with placental dysfunction since gross abnormalities of the placenta and umbilical cord are diagnostic features of maternal vascular malperfusion (MVM) [10,11,12,13,14,15,16,17]. However, data on the predictive accuracy of these latter sonographic placental markers, and whether they are predictive of placental disease independent of uterine artery Doppler, are scarce and vary by the inherent risk in studied populations. Furthermore, the characteristics of the association between measure of placental size and placental dysfunction are currently unclear, and the cutoff values used to interpret some of these markers vary between studies and require further justification and validation [12,13].

In the current study, we aimed to characterize the association of second-trimester sonographic placental morphology markers with placenta-mediated complications, and determine whether these markers are predictive of placental dysfunction independent of uterine artery Doppler.

## 2. Methods

### 2.1. Study Design and Participants

This was a retrospective cohort study of all patients with a singleton pregnancy at high risk of preeclampsia [18] or fetal growth restriction who underwent a sonographic placental study at mid-gestation (16^0/7^–24^6/7^ weeks’ gestation) in a single tertiary referral center (Sunnybrook Health Sciences Centre, Toronto, Ontario) between June 2016 and March 2019. Patients with any of the following conditions were excluded from the study: uncertain gestational dating based on first-trimester ultrasound, birth prior to 20 weeks of gestation, known genetic or structural fetal anomalies, or missing data on sonographic placental assessment or on pregnancy outcomes. The study was approved by the Sunnybrook Health Sciences Center Research Ethics Board.

### 2.2. Data Collection

Patients were identified through the institutional ultrasound database. Medical charts and ultrasound reports of eligible patients were reviewed for demographic data, medical history, sonographic placental study results, antenatal complications, gestational age at birth, and birthweight.

### 2.3. Sonographic Placental Markers

According to our departmental protocols, all patients determined to be at high risk of placenta-mediated complications undergo a sonographic placental study at 16–24 weeks’ gestation by an experienced team of sonographers that underwent standardized training in placental assessment as previously described [19]. A commercially available Voluson E8/E10 Expert machine (GE Healthcare, Kretz Ultrasound, Zipf, Austria) equipped with an abdominal C-2-9D XDclear Convex Array Probe and RAB6-D Ultralight Real Time 4D Convex Array Probe. Briefly, the sonographic placental study includes assessment of: (1) Placental dimensions (**length**, **width**, and **thickness,** as illustrated in Figure 1) in the absence of a uterine contraction; (2) **Placental appearance** (subjective description of placental texture appearance as homogeneous vs. heterogeneous, the presence of echogenic placental lesions [14], and the presence of echo-dense areas); (3) **Placental cord insertion site** (classified as velamentous, marginal (<2 cm from placental edge), or central); (4) **Number of vessels in cord** (i.e., 3- vs. 2-vessel cord); and (5) **Uterine artery Doppler**, described both as mean pulsatility index (PI) and the presence of bilateral early diastolic notching. For the purpose of the current analysis, we calculated two additional markers: **placental area** (estimated using the product of placental length × width) and **placental volume** (estimated using the product of placental length × width × thickness). Uterine artery Doppler evaluation (either trans-abdominally or trans-vaginally) was carried out as per ISUOG guidelines [18].

### 2.4. Outcomes

The primary outcome was a composite variable of placenta-mediated complications including either early-onset preeclampsia (defined as preeclampsia requiring delivery prior to 34 weeks’ gestation) or fetal growth restriction (defined as birthweight below the 3rd percentile for gestational age according to the Hadlock standard [20]). Preeclampsia was diagnosed according to the Task Force on Hypertension in Pregnancy recommendations [21].

### 2.5. Data Analysis

Baseline characteristics, distribution of sonographic placental markers, and outcomes were compared between patients who developed the primary outcome and those who did not. The student’s t-test was used for continuous variables and the chi-square test or the Fisher’s exact test were used for categorical variables, as appropriate.

Given that the distribution of some of the placental markers (placental length, width, thickness, area, volume, and mean uterine artery PI) vary with gestational age, these markers were expressed as multiple of median (MoM). Median values at each week were derived from our entire cohort (see Appendix A), as there are currently no week-specific reference values for these markers for the gestational age range investigated in the current study. For uterine artery Doppler PI, we also considered the crude (non-transformed) PI values as a dichotomous variable in relation to the 95th percentile of a previously published reference [22].

The association of each marker with the primary outcome was determined using logistic regression analysis and was expressed as an adjusted odds ratio (aOR) with 95%-confidence interval (95%-CI). Models were adjusted for the baseline characteristics that differed between patients with and without the primary outcome (Model 1). In addition, to identify markers that are associated with the primary outcome independent of the other placental markers, we developed a second set of models that were also adjusted for the other placental markers (Model 2).

Receiver-operator characteristic (ROC) analysis was used to determine the area under the ROC curve (AUC) as a threshold-independent overall measure of the discriminative ability of each placental marker. The AUC of the different markers were compared using the method of Hanley and McNeil [23]. The ROC analysis was also used to identify predefined thresholds for each of the continuous placental markers, including the threshold associated with a sensitivity of 80% and the threshold associated with a specificity of 80%. These thresholds were used to facilitate comparison of the sensitivity (for a fixed specificity) and the specificity (for a fixed sensitivity) of the different markers.

The predictive accuracy of each marker in isolation and in combination with other markers was described using the following measures: sensitivity, specificity, positive and negative predictive value (PPV and NPV, respectively), overall accuracy (defined as the proportion of true results (either true positive or true negative), and positive and negative likelihood ratio (LR+ and LR−, respectively). LR+ > 10 and LR− < 0.1 were considered to provide strong predictive value; LR+ of 5–10 and LR− of 0.1–0.2 were considered to reflect moderate predictive value; and LR+ < 5 and LR− > 0.2 were considered to reflect only low predictive value [24,25].

Data were analyzed using the SPSS statistical software Version 25.0 (IBM Corp., Armonk, NY, USA). Significance was set to a two-sided *p*-Value of <0.05.

## 3. Results

### 3.1. Characteristics and Outcomes of the Study Groups

Of the total 429 eligible patients identified during the study period, 45 (10.5%) experienced the primary outcome of placenta-mediated complication. Patients who experienced a placenta-mediated complication were more likely to have pre-existing hypertension or a history of preeclampsia and fetal growth restriction in a prior pregnancy (Table 1).

Patients in the placenta-mediated complications group had a higher rate preeclampsia (both at term and preterm) and preterm birth, and had a lower mean birthweight compared with those without placenta-mediated complications (Table 1).

### 3.2. Rate and Distribution of Sonographic Placental Markers

As the first step, to identify the sonographic placental markers that were most informative with respect to the risk of placenta-mediated complications, we compared the distribution of these markers between patients who did vs. did not experience a placenta-mediated complication (Table 2). Patients who experienced placenta-mediated complications were characterized by a significantly shorter mean placental length and width, a smaller placental area and volume, a higher mean uterine artery PI, and a higher rate of uterine artery PI > 95th centile and of bilateral uterine notching compared with patients who did not experienced placenta-mediated complications (Table 2). There were no differences between the groups in mean placental thickness, the rate of abnormal placental morphology, the rate of incidence of a 2-vessel cord, or the rate of marginal or velamentous placental cord insertion (Table 2).

To better characterize the relationship between the continuous placental markers and the risk of placenta-mediated complications, the rate of the primary outcome was stratified by quartiles of each of the continuous markers (Figure 2). The rate of the primary outcome increased progressively with decreasing placental length, width, and area, and increased progressively with increasing mean uterine artery PI. By contrast, the relationship of placental thickness with the primary outcome followed a U-shaped pattern, where the rate of the primary outcome was highest for the lowest and highest quartiles of placental thickness (Figure 2). This pattern is illustrated even more clearly when the rate of the primary outcome is shown in relation to deciles of placental thickness. Given this U-shaped pattern, we considered an alternative placental thickness variable that would identify placentas that are either too thick or too thin (calculated as the absolute value of [1 minus placental thickness (in MoM)]), but even this variable did not differ between patients who did vs. did not experience the primary outcome (Table 2). For this reason, we did not consider placental thickness and placental volume (which was calculated based on placental thickness) in the subsequent analysis.

### 3.3. Association of Sonographic Placental Markers with Adverse Outcomes

The associations of the sonographic placental markers with the primary outcome are presented in Table 3. Placental length, placental width, placental area, mean uterine artery PI and bilateral uterine artery notching were all associated with the primary outcome, even when the models were adjusted for baseline characteristics that differed between patients with and without the primary outcome (Table 3, Model 1). Placental area (which combines both placental length and width) and mean uterine artery PI were each independently associated with the primary outcome even when the models were adjusted for placental area, mean uterine artery and bilateral uterine artery notching (aOR 0.21 (95%-CI 0.06–0.73) and aOR 11.71 (95%-CI 3.84–35.72), respectively) (Table 3, Model 2). In contrast, bilateral uterine artery notching was not associated with the primary outcome when the model was adjusted for placental area and mean uterine artery PI. For mean uterine artery PI, associations were stronger for the outcome of early-onset preeclampsia than for the outcome of birthweight < 3rd centile (aOR 29.88 (95%-CI 5.90–151.32) vs. aOR 8.42 (95%-CI 3.95–17.98)) (Table 3).

### 3.4. Predictive Accuracy of Sonographic Placental Markers for Adverse Outcomes

The ROC curves of the sonographic placental markers for the prediction of the primary outcome are shown in Figure 3. The AUC was highest for mean uterine artery PI (0.80), which was significantly higher that the AUC of placental length (0.65, *p* = 0.020) or placental area (0.67, *p* = 0.044). A model that included both mean uterine artery PI and placental area produced a small non-significant increase in the AUC (0.82; *p* = 0.255 for comparison with mean uterine artery PI) (Figure 3).

We finally calculated the predictive accuracy of placental area, mean uterine artery PI, and a model that combines these two markers, for the primary outcome (Table 4). For a fixed sensitivity of 80%, the mean uterine artery PI had a higher specificity for the primary outcome compared with placental area (52% [95%-CI 47–57%] vs. 39% [95%-CI 34–45%]), and the combination of both markers was associated with a relatively minor increase in specificity for the primary outcome (63% [95%-CI 58–68%]). For a fixed specificity of 80%, there was no statistically significant difference in the sensitivity of placental area (43% [95%-CI 27–61%]), mean uterine artery PI (63% [95%-CI 47–77%]), or the combination of both markers (65% [95%-CI 48–80%]) (Table 4).

## 4. Discussion

### 4.1. Principal Findings

The aim of the current study was to characterize the association of second-trimester sonographic placental markers with adverse clinical outcomes that are strongly associated with placental dysfunction, and to determine if these markers are predictive of placental dysfunction independent of uterine artery Doppler waveform assessment. Our main findings were as follows: (1) Mean uterine artery Doppler PI exhibited a strong association with the risk of placental complications; the presence of bilateral uterine notching was not associated with placental complications once the analysis was adjusted for mean uterine artery PI; (2) Placental size (as reflected by either placental length, width, or area) demonstrated an inverse continuous relationship with these complications; (3) Placental area was associated with placental complications independent of uterine artery Doppler, but the addition of placental area to mean uterine artery PI did not result in any significant improvement in screening accuracy; (4) Placental thickness demonstrated a U-shaped relationship with the risk of placental complications.

### 4.2. Interpretation of the Results in the Context of Previous Observations

Measures of placental size have been shown to be associated with placenta-mediated complications [10,13]. For example, the maximal curved linear length along the maternal interface was found to be shorter in placentas with evidence of maternal vascular malperfusion pathology (0.98 ± 0.17 vs. 1.03 ± 0.16 MoM; *p* = 0.03); the AUC for the prediction of placental complications was 0.68, which is similar to our findings (0.67). Of note, one study [13] was limited to healthy nulliparous patients and used a less severe primary outcome (any preeclampsia or birthweight < 10th centile) than the one used in the current study.

Data on the association of placental thickness with placenta-mediated complications are conflicting, in part as several pathologic processes may increase placental thickness [11]. Several clinical studies identified an association between an abnormally thick placenta and placenta-mediated complications, although the definition of a thick placenta varied widely between studies and included either a subjectively thickened appearance [26], a maximum thickness of >4 cm [16,27], thickness > 90th centile [28] or >95th centile for gestational age [29,30], thickness > 1.2 MoM [31], or thickness > 50% of length [27]. However, at the same time, placental complications have also been reported to be associated with abnormally thin placentas [32,33]. These observations likely explain our finding of a U-shaped relationship between placental thickness and the risk of placental complications. Our study design did not allow us to evaluate the underlying causes of increased placental thickness, which are reported with various fetal anomalies, congenital infections, hydrops, maternal diabetes, chronic placental abruption, or Breus mole [28,30,33].

Data on the association of placental volume with placenta-mediated complications have been conflicting as well. While some have shown that placental volume, when measured using 3-dimensional sonography, can identify pregnancies at risk of placental dysfunction [34,35,36], others reported that placental volume alone was of limited value in predicting placental complications [37,38,39]. We speculate that this controversy may also be attributed, at least in part, to the complex relationship between placental thickness (a determinant of placental volume) and placental complications.

Whether the association of placental size with placental complications is independent of uterine artery Doppler remains unclear. Smaller placental size in combination with high uterine artery PI at 11–13 weeks’ gestation had a better predictive accuracy for preeclampsia and fetal growth restriction compared to each parameter in isolation [31,36,40]. We previously demonstrated that, in patients with unexplained elevated second-trimester biomarkers, a thick placenta (defined as a maximum thickness of >4 cm or >50% of placental length) and abnormal uterine artery Doppler had a stronger association with placenta-mediated complications when combined compared to each marker in isolation [27]. In the current study, the contribution of placental area had a small non-significant contribution to the predictive accuracy for the primary outcome. This observation may be explained by the fact that in the current study, rather than interpreting placental markers in a dichotomous manner, we considered these markers as continuous variables and evaluated their predictive accuracy for severe placenta-mediated complications as opposed to milder outcomes used in previous studies [27].

### 4.3. Strengths and Limitations

The main strengths of the current study are the relatively large sample size, the detailed information on placental markers, and the fact that all patients were screened and managed by an experienced maternal-fetal medicine team adopting a standardized management protocol within a single center. Another strength is the use of a clinically relevant primary outcome of severe placenta-mediated complications as opposed to less severe outcomes (such as any preeclampsia or birthweight < 10th centile) used by previous studies that have a weaker association with the gross findings in placentas that are affected by maternal vascular malperfusion disease.

The main limitations of the current study are its retrospective design and the limited information on other, non-sonographic biomarkers for placental dysfunction, such as maternal blood pressure and circulating angiogenic proteins [41,42]. Low circulating PlGF is strongly associated with preterm delivery associated with placental pathology [43]. In a recent publication from our group involving 979 subjects, low circulating placenta growth factor (PlGF) prior to 24 weeks gestational age was strongly associated with stillbirth [44]. Another limitation is that the median values for measures of placental size (length, width, thickness, and area) were derived from the current cohort of patients at an increased risk of placental dysfunction, due to the lack of week-specific reference values for these measures in low-risk uncomplicated pregnancies for the entire gestational age range investigated in the current study. In addition, in the current study, placental area and volume were grossly estimated as the product of the unidimensional placental measures (length, width, and thickness) rather than measured directly by ultrasound. While placental area could have been estimated more accurately by using an equation for an area of an ellipse (PI × width/2 × length/2), such a simple transformation (multiplication by a constant value) would not have affected the association or predictive accuracy estimates for these markers. Finally, our study was underpowered to study uncommon placenta-mediated complications, such as stillbirth and placental abruption.

## 5. Conclusions

In summary, we found that mean uterine artery PI is the strongest predictor for placenta-mediated complications during 16 to 24 weeks’ gestation. Placental area was also associated with such complications yet, when combined with uterine artery Doppler, did not significantly improve the predictive accuracy for placenta-mediated complications when compared with uterine artery PI in isolation. Prospective studies are needed which combine these imaging modalities with circulating angiogenic proteins, in particular placenta growth factor, to determine the most cost-effective approach to risk stratification of care for pregnant individuals considered to be at risk of placental dysfunction disorders.

## Figures and Tables

**Figure 1 jcm-11-06759-f001:**
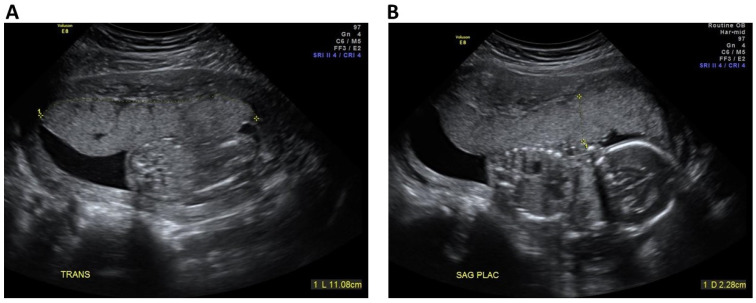
Illustration of measurement of placental dimensions. The figures illustrate the measurement of placental length (**A**) and thickness (**B**) in patient undergoing placental assessment at 18 weeks’ gestation. Following initial axial and sagittal placental scan, the maximal placental length using curved linear method at the basal plate, and maximal thickness.

**Figure 2 jcm-11-06759-f002:**
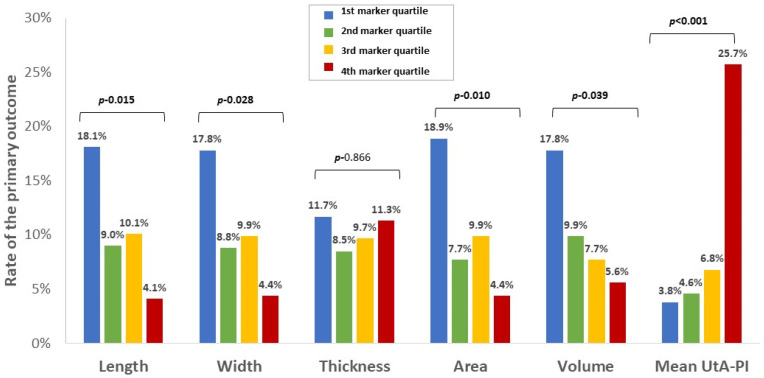
Relationship between the continuous placental markers and rate of the primary outcome. The rate of the primary outcome is shown for each quartile of the six continuous sonographic placental markers (expressed in MoM): placental length, width, thickness, area, and volume, and uterine artery PI. MoM, multiples of median; PI, pulsatility index; UtA, uterine artery.

**Figure 3 jcm-11-06759-f003:**
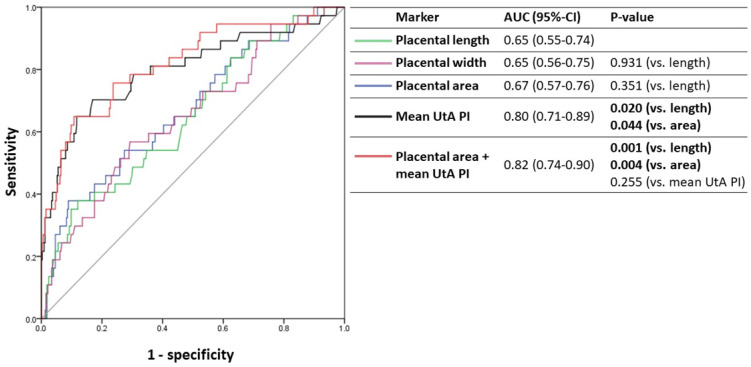
ROC curves of the sonographic placental markers for the prediction of placental complications. The ROC curves for the prediction of the primary outcome are shown for each of the following placental markers (expressed in MoM): placental length (green line), placental width (purple line), placental area (blue line), mean uterine artery PI (black line), and a combination of placental area and mean uterine artery PI (red line). AUC, area under the curve; ROC, receiver operating characteristic; MoM, multiples of median; PI, pulsatility index; UtA, uterine artery.

**Table 1 jcm-11-06759-t001:** Baseline characteristics and pregnancy outcomes.

Characteristic/Outcome	Overall Cohort *n* = 429	Placenta Mediated Complications ^a^ *n* = 45	No Placenta Mediated Complications *n* = 384	*p*-Value
Maternal age (years), *mean ± SD*	34.0 ± 4.9	33.5 ± 6.0	34.0 ± 4.8	0.601
>35 years, *n (%)*	160 (37.3)	18 (40.0)	142 (37.0)	0.745
Pre-pregnancy BMI (Kg/m^2^), *mean ± SD*	26.3 ± 6.4	26.4 ± 6.1	26.3 ± 6.4	0.937
Nulliparity, *n (%)*	147 (34.3)	15 (33.3)	132 (34.4)	1.000
* **Maternal co-morbidity** *				
Pre-existing hypertension, *n (%)*	37 (8.6)	9 (20.0)	28 (7.3)	**0.009**
Diabetes mellitus, *n (%)*	9 (2.1)	2 (4.4)	7 (1.8)	0.259
* **Obstetrical history** *				
Past gestational diabetes, *n (%)*	20 (4.7)	1 (2.2)	19 (4.9)	0.709
Past preeclampsia, *n (%)*	60 (14.0)	11 (24.4)	49 (12.8)	**0.041**
Past preterm birth, *n (%)*	71 (16.6)	9 (20.0)	62 (16.1)	0.525
Past placental abruption, *n (%)*	19 (4.4)	1 (2.2)	18 (4.7)	0.707
Past fetal growth restriction, *n (%)*	59 (13.8)	12 (26.7)	47 (12.2)	**0.019**
Past stillbirth, *n (%)*	48 (11.2)	6 (13.3)	42 (10.9)	0.618
GA at placenta study (weeks), *mean ± SD*	20.8 ± 2.3	20.8 ± 2.1	20.8 ± 2.3	0.292
* **Pregnancy outcomes** *				
Gestational diabetes mellitus, *n (%)*	49 (11.4)	3 (6.7)	46 (12.0)	0.455
Placental abruption, *n (%)*	13 (3.0)	3 (6.7)	10 (2.6)	0.146
Preeclampsia, *n (%)*	56 (13.1)	15 (33.3)	41 (10.7)	**<0.001**
Requiring delivery < 37 weeks, *n (%)*	18 (4.2)	12 (26.7)	6 (1.6)	**<0.001**
Requiring delivery < 34 weeks, *n (%)*	9 (2.1)	9 (20.0)	0 (0)	**<0.001**
GA at birth (weeks), *mean ± SD*	37.7 ± 3.3	33.2 ± 6.2	37.8 ± 2.3	**<0.001**
<37 weeks, *n (%)*	68 (15.9)	21 (46.7)	47 (12.2)	**<0.001**
<34 weeks, *n (%)*	31 (7.2)	18 (40.0)	13 (3.4)	**<0.001**
Birth weight (g), *mean ± SD*	2956 ± 740	1657 ± 839	3104 ± 561	**<0.001**
Birth weight < 3rd centile, *n (%)*	42 (9.8)	42 (93.3)	0 (0)	**<0.001**
Female neonate, *n (%)*	221 (5.1)	27 (61.4)	194 (50.5)	0.203

BMI, body mass index; SD, standard deviation; GA, gestational age. ^a^ Defined as preterm preeclampsia or birthweight < 3rd centile. Significant *p*-values are emphasized in bold font.

**Table 2 jcm-11-06759-t002:** Distribution of the sonographic markers in patients with and without placenta-mediated complications.

Placenta Sonographic Marker	Overall Cohort *n* = 429	Placenta Mediated Complications ^a^ *n* = 45	No Placenta Mediated Complications *n* = 384	*p*-Value
Placental length (MoM), *mean ± SD*	0.99 ± 0.20	0.89 ± 0.16	1.00 ± 0.20	**0.001**
Placental width (MoM), *mean ± SD*	1.08 ± 0.26	0.96 ± 0.22	1.09 ± 0.26	**0.003**
Placental thickness (MoM), *mean ± SD*	1.05 ± 0.31	1.04 ± 0.31	1.05 ± 0.31	0.756
Placental absolute (1-thickness [MoM]), *mean ± SD*	0.24 ± 0.21	0.25 ± 0.18	0.24 ± 0.21	0.637
Placental area (MoM), *mean ± SD*	1.08 ± 0.43	0.87 ± 0.34	1.10 ± 0.43	**0.002**
Placental volume (MoM), *mean ± SD*	1.14 ± 0.60	0.89 ± 0.42	1.17 ± 0.61	**0.007**
Abnormal placental morphology, *n (%)*	67 (15.6)	10 (22.2)	57 (14.8)	0.196
2-vessel cord, *n (%)*	8 (1.9)	2 (4.4)	6 (1.6)	0.201
Marginal/velamentous cord insertion, *n (%)*	40 (9.3)	8 (17.8)	32 (8.3)	0.054
Mean uterine artery PI (MoM), *mean ± SD*	1.09 ± 0.42	1.56 ± 0.62	1.04 ± 0.30	**<0.001**
Mean uterine artery PI > 95th %, *n (%)*	48 (11.2)	19 (49.2)	29 (7.6)	**<0.001**
Bilateral uterine artery notching, *n (%)*	39 (9.1)	16 (35.6)	23 (6.0)	**<0.001**

MoM, multiples of median; PI, pulsatility index. ^a^ Defined as early-onset preeclampsia or birthweight < 3rd centile. Significant *p*-values are emphasized in bold font.

**Table 3 jcm-11-06759-t003:** Association of placental sonographic markers with placenta-mediated complications.

Outcome	Placental Marker	Crude OR (95% CI)	Adjusted OR (95% CI)—Model 1 ^b^	Adjusted OR (95% CI)—Model 2 ^c^
Placenta-mediated complications ^a^	Length (MoM)	**0.05 (0.01–0.31)**	**0.05 (0.01–0.33)**	**-**
	Width (MoM)	**0.11 (0.03–0.47)**	**0.09 (0.02–0.42)**	**-**
	Area (MoM)	**0.19 (0.06–0.55)**	**0.17 (0.06–0.54)**	**0.21 (0.06–0.73)**
	Bilateral notching	**8.66 (4.12–18.18)**	**6.77 (3.09–14.83)**	1.59 (0.49–5.11)
	Mean UtA PI (MoM)	**14.61 (6.66–32.05)**	**12.49 (5.61–27.80)**	**11.71 (3.84–35.72)**
Preeclampsia < 34 weeks	Length (MoM)	0.07 (0.01–1.65)	0.13 (0.01–2.96)	-
	Width (MoM)	0.09 (0.01–1.28)	0.06 (0.01–1.22)	-
	Area (MoM)	0.17 (0.21–1.36)	0.20 (0.02–1.65)	0.59 (0.06–5.74)
	Bilateral notching	**23.46 (5.61–98.08)**	**14.04 (2.82–70.03)**	1.15 (0.10–13.01)
	Mean UtA PI (MoM)	**33.79 (8.87–128.70)**	**29.88 (5.90–151.32)**	**22.91 (3.21–163.55)**
Birthweight < 3rd centile	Length (MoM)	**0.06 (0.01–0.36)**	**0.06 (0.01–0.38)**	**-**
	Width (MoM)	**0.11 (0.03–0.52)**	**0.10 (0.02–0.49)**	**-**
	Area (MoM)	**0.20 (0.07–0.60)**	**0.19 (0.06–0.60)**	**0.26 (0.08–0.86)**
	Bilateral notching	**6.22 (2.89–13.85)**	**5.09 (2.26–11.49)**	1.32 (0.40–4.41)
	Mean UtA PI (MoM)	**9.38 (4.50–19.55)**	**8.42 (3.95–17.98)**	**7.56 (2.63–21.67)**

MoM, multiples of median; UtA, uterine artery; PI, pulsatility index. ^a^ Defined as early-onset preeclampsia or birthweight < 3rd centile. ^b^ Models adjusted for chronic hypertension and a history of preeclampsia or fetal growth restriction in prior pregnancy. ^c^ Models adjusted for the same variables included in Model 1, as well as for the following other placental markers (for placental area—model was adjusted for bilateral uterine artery notching and mean uterine artery PI (MoM); for bilateral UtA notching—model was adjusted for placental area (MoM) and mean uterine artery PI (MoM); for mean UtA PI—model was adjusted for placental area (MoM) and bilateral uterine artery notching). These models were not calculated for placental length and placental width due to the correlation between these markers and placental area, which resulted in multicollinearity. Significant associations are emphasized in bold font.

**Table 4 jcm-11-06759-t004:** Predictive accuracy of placental area and uterine artery for the primary outcome.

Threshold Type	Marker	Threshold	Test Positive Rate [n (%)]	Sens. (%) (95%-CI)	Spec. (%) (95%-CI)	PPV (%) (95%-CI)	NPV (%) (95%-CI)	Accuracy (%) (95%-CI)	+LR (95%-CI)	−LR (95%-CI)
Sensitivity of 80%	Placental area (MoM)	<1.1158	30 (81.1)	81 (65–92)	39 (34–45)	13 (11–15)	95 (90–97)	44 (39–49)	1.3 (1.1–1.6)	0.5 (0.2–0.9)
	Mean UtA PI (MoM)	>0.9830	35 (81.4)	81 (67–92)	52 (47–57)	16 (14–19)	96 (93–98)	55 (50–60)	1.7 (1.4–2.0)	0.4 (0.2–0.7)
	Placental area *and* mean UtA PI	>0.0596	30 (81.1)	81 (65–92)	63 (58–68)	20 (17–24)	97 (94–98)	65 (60–70)	2.2 (1.8–2.7)	0.3 (0.1–0.6)
Specificity of 80%	Placental area (MoM)	<0.7585	16 (43.2)	43 (27–61)	80 (75–84)	20 (14–27)	96 (90–94)	96 (72–81)	2.2 (1.4–3.3)	0.7 (0.5–0.9)
	Mean UtA PI (MoM)	>1.2386	27 (62.8)	63 (47–77)	80 (76–84)	27 (21–33)	95 (93–97)	77 (74–82)	3.2 (2.3–4.3)	0.5 (0.3–0.7)
	Placental area *and* mean UtA PI	>0.1055	24 (64.9)	65 (48–80)	80 (75–84)	27 (21–34)	95 (93–97)	79 (74–83)	3.2 (2.4–4.5)	0.4 (0.3–0.7)

UtA, uterine artery Doppler; MoM, multiples of median; PI, pulsatility index; Sens., sensitivity; Spec., specificity; PPV, positive predictive value; NPV, negative predictive value; +LR, positive likelihood ratio; −LR, negative likelihood ratio.

## Data Availability

Data available on request due to privacy and ethical restrictions.

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
