# Peer review of "Contribution of Second Trimester Sonographic Placental Morphology to Uterine Artery Doppler in the Prediction of Placenta-Mediated Pregnancy Complications"

_jcm, 2022, doi:10.3390/jcm11226759_

Round 1

Reviewer 1 Report

Thank you for the opportunity to review the manuscript »Contribution of Second Trimester Sonographic Placental Morphology to Uterine Artery Doppler in the Prediction of Placenta-Mediated Pregnancy Complications. «The retrospective cohort study aimed to evaluate the association of placental biometric and morphological as well as uterine arteries doppler characteristics with placenta-related complications, specifically preeclampsia and fetal growth restriction < 3rd centile. The study's results have shown that between the 16th and 24th gestational weeks, UtA-mPI is the best predictive value for combined and individual placenta-related complications. Among placental biometric values, only the placental area was significantly associated with placenta-related complications; however, it does not add significantly to the prediction when combined with UtA-mPI.

The manuscript is well-written and easy to read. The subject of the study is not novel. However, the clearly stated aim and appropriate design of the study add to the clinical knowledge. The abstract and the tweetable abstract provide an accessible summary of the paper. The keywords accurately reflect the content. The introduction appropriately sets out the argument and demonstrates the need for the investigation. The aim of the study is precise. The statistics are well done, and the results are presented clearly. The tables and figures complement the results meaningfully, and they are adequately presented. The discussion gathers all results together and supports their findings well with the correlation with other studies. The limitations of the study are appropriately presented. The results support the conclusion. The literature is relevant to the subject of the study.

Some comments:

Methods

2.1. Study design and participants

1. Please, describe or put a reference in the manuscript which screening method or protocol was used to determine the risk of preeclampsia or fetal growth restriction in the cohort (Task Force, NICE, FMF algorithm…)? 

2. Were any patients treated with prophylactic low-dose aspirin or antihypertensive drugs at the US examination? If so, this data should also be considered in the analyses.

2.3. Sonographic placental markers

3. Which US device(s) and probe(s) were used for examination and doppler measurements?

4. Please, describe a technique of the sonographic placental measurements so that the measurements can be reproduced (trace…, where markers were placed…, where was measured thickness.).

5. How were fundal placentas measured?

6. Please, describe the technique of UtA doppler measurements.

Conclusions

7. It should be mentioned that the conclusions refer to the 16th – 24th week of pregnancy.

References

8. References should be arranged uniformly.

9. References 16 and 29 and 18 and 44 are the same.

Reviewer 2 Report

The authors investigated whether sonographic assessment of the placenta and umbilical cord improves placental dysfunction diagnosis compared to uterine artery Doppler waveform assessment alone. The study found that while sonographic placental area is a predictor for placenta-mediated complications, it did not significantly improve the prediction accuracy for complications when used in combination with uterine artery PI compared with uterine artery PI in isolation. Overall, the manuscript is well-written, and I have a few minor comments:

 Abstract – The font for the conclusion appears to be different to the rest of the text, please correct.

Introduction – All of the references throughout the introduction are not superscripted, please correct.

Results - Tables 1 and 2; data for Placenta and No placenta mediated complications have +-, which should be changed to ±.

The authors refer to supplemental material, but none was included in the uploaded manuscript.

The following sections were not filled out: Author Contributions, Funding, Institutional Review Board Statement, Informed Consent Statement, Data Availability Statement, Conflicts of Interest (where applicable).
